# Predicting Outcome after Percutaneous Ablation for Early-Stage Hepatocellular Carcinoma Using Various Imaging Modalities

**DOI:** 10.3390/diagnostics13193058

**Published:** 2023-09-26

**Authors:** Ryo Shimizu, Yoshiyuki Ida, Masayuki Kitano

**Affiliations:** Second Department of Internal Medicine, Wakayama Medical University, 811-1 Kimiidera, Wakayama 641-8509, Japan

**Keywords:** hepatocellular carcinoma, predicting outcome, ablation, ultrasonography (US), computed tomography (CT), magnetic resonance imaging (MRI), 18F-fluorodeoxyglucose positron emission tomography (18F-FDG PET)

## Abstract

Percutaneous ablation is a low-invasive, repeatable, and curative local treatment that is now recommended for early-stage hepatocellular carcinoma (HCC) that is not suitable for surgical resection. Poorly differentiated HCC has high-grade malignancy potential. Microvascular invasion is frequently seen, even in tumors smaller than 3 cm in diameter, and prognosis is poor after percutaneous ablation. Biopsy has a high risk of complications such as bleeding and dissemination; therefore, it has limitations in determining HCC tumor malignancy prior to treatment. Advances in diagnostic imaging have enabled non-invasive diagnosis of tumor malignancy. We describe the usefulness of ultrasonography, computed tomography, magnetic resonance imaging, and 18F-fluorodeoxyglucose positron emission tomography for predicting outcome after percutaneous ablation for HCC.

## 1. Introduction

Percutaneous ablation techniques such as radiofrequency ablation (RFA) and microwave ablation have been established as minimally invasive, repeatable, and curative treatments for small hepatocellular carcinoma (HCC) and are recommended for early-stage HCC in the various clinical practice guidelines for the management of HCC [1,2,3]. In these guidelines, the indication for ablation is three nodules or less of diameter ≤3 cm. However, several studies have reported rapid progression such as early local recurrence, seeding, intrahepatic dissemination, advanced recurrence with vessel invasion, and/or extrahepatic metastasis after RFA [4,5,6,7,8,9,10]. The post-RFA prognosis is particularly dismal for poorly differentiated HCC [11] due to the high risk of neoplastic seeding [12] and early local tumor recurrence within one year [13]. Furthermore, the overall survival rate after RFA for HCC had been shown to worsen significantly with increasing histopathological grade (Edmondson–Steiner grade) [11]. Because the frequency of vascular invasion and intrahepatic metastasis increases as the degree of differentiation decreases, even in small tumors (≤3 cm in diameter) [14,15,16,17], even small HCCs that appear suitable candidates for ablation should be assessed for histological differentiation to determine the optimal treatment plan for each patient. However, the feasibility of biopsy has limitations according to the tumor location and the risk of complications such as bleeding and needle trace seeding [18,19,20,21]. In addition, HCC may be heterogeneous in histology, and biopsy may not be able to diagnose the degree of histological differentiation accurately. Therefore, the development of non-invasive imaging methods for predicting prognosis after ablation for HCC is highly desirable.

We describe the utility of ultrasonography (US), computed tomography (CT), magnetic resonance imaging (MRI), and 18F-fluorodeoxyglucose positron emission tomography (18F-FDG PET) for predicting outcome after ablation for HCC.

## 2. Ultrasonography

US currently offers the highest spatial resolution of any imaging modality used to diagnose HCC. However, some HCCs have indistinct boundaries and are often difficult to evaluate by B-mode US. In normal hepatocytes, most blood flow originates from the portal vein, whereas in HCC, blood flow from the arteries is dominant. Contrast-enhanced US (CEUS) takes advantage of this blood flow characteristic. Furthermore, because contrast agents for ultrasound contain microbubbles, CEUS can be safely performed in patients who cannot undergo contrast-enhanced CT or MRI due to renal dysfunction, iodine allergy, or asthma. Four ultrasound contrast agents are currently available internationally for use in liver imaging [22]:-Definity^®^/Luminity^®^ (Lantheus Medical Imaging Inc., North Billerica, MA, USA);-SonoVue^®^/Lumason^®^ (Bracco Suisse SA, Geneva, Switzerland);-Optison^®^ (GE Healthcare AS, Oslo, Norway);-Sonazoid^®^ (GE Healthcare AS, Oslo, Norway).

CEUS of the liver is performed in three phases: arterial phase, portal venous phase, and late phase. Among the four agents, only Sonazoid^®^ can be used to observe the post-vascular phase (PVP) [22]. Sonazoid^®^ contains perfluorobutane microbubbles and was first launched in Japan in January 2007. It is unique in that it is phagocytosed by Kupffer cells following the vascular phase of imaging [23]. Compared with normal hepatocytes, malignant tumors contain fewer or absent Kupffer cells. Accordingly, malignant tumors are depicted as image defects in the PVP. Numerous reports have attempted to use ultrasound contrast agents to assess the gross type of HCC lesions, as well as blood flow, to determine malignancy.

### 2.1. Gross Classification

The gross classification of HCC is divided five types [24]:-The small nodular type with indistinct margin.-The simple nodular (SN) type, with a clear round shape.-The simple nodular with extranodular growth (SNEG) type, with a clear round shape and one or more tumor growths coexisting around the nodules.-The confluent multinodular (CMN) type, consists of clusters of small nodules.-The infiltrative type.

Malignant potential is also classified on the basis of gross pathologic type. Among the types, the non-SN type (including SNEG and CMN) is a useful predictive factor of microvascular invasion (MVI) pathologically and has a higher frequency of vessel invasion compared with the SN type. Therefore, the non-SN type is considered to have a higher grade of malignancy than the SN type [16,25,26,27] (Table 1).

Previous studies have observed the malignant characteristics of each gross HCC type using Sonazoid^®^ in the PVP. Hatanaka et al. reported that the gross types of HCC shown in the PVP by CEUS with Sonazoid^®^ were useful for evaluating the malignancy of these tumors. They classified tumors in the PVP images of CEUS performed before liver resection into SN and non-SN types, and matched the images to the tumor morphology of the resected specimens [28].

In their study based on postoperative pathological reports, Li et al. reported that preoperative PVP images obtained with CEUS using Sonazoid^®^ can predict the risk of MVI [29]. Nuta et al. observed tumor contours with B-mode and in the PVP with Sonazoid^®^, classified the tumors into irregular and non-irregular types, and investigated their pathologic grade and prognosis. They defined irregular type as HCCs that had an irregular or unclear contour on B-mode or an irregular defect on PVP images with CEUS (Figure 1); all images that were not irregular-type HCC were categorized as non-irregular type (Figure 2). They reported predictability of clinicopathological type (irregular type would be non-SN type, non-irregular type would be SN type) using PVP CEUS images was better than using conventional B-mode images. Moreover, analysis of PVP images revealed cumulative 5-year rates of three or more recurrences of 43% and 7% in the irregular-defect and non-irregular-defect groups, respectively (*p* = 0.028) [30].

### 2.2. Intratumoral Blood Flow

Intratumoral blood flow in HCC differs depending on the degree of differentiation. It has been reported that, as HCC grows and becomes more dedifferentiated, the number of intratumoral arteries increases while the number of portal veins decreases. In addition, the major drainage vessel in HCC switches from hepatic veins in well-differentiated tumors to hepatic sinuses in the moderately differentiated and then to portal veins. Therefore, portal blood flow changes significantly during the course of liver carcinogenesis. In contrast, the density of intranodal hepatic veins decreases significantly as the HCC progresses from well to moderate differentiation [31]. Therefore, observation of tumor blood flow by CEUS is also useful in predicting malignancy in HCC. Maruyama et al. performed preoperative measurement of the difference in signal intensity between HCC and liver parenchyma 4 s after Sonazoid^®^ reached the hepatic artery and examined its relationship to prognosis after RFA. In their prospective study, the cumulative distant recurrence rate for patients with intensity differences greater than 10 dB and less than 10 dB in early arteries was 23.9% and 33.3% at 1 year and 65.1% and 91.3% at 2 years, respectively. [32]. Han et al. reported that peak intensity and time to peak in the arterial phase of CEUS with SonoVue^®^ before RFA can predict intrahepatic recurrence after treatment [33].

Several studies have evaluated the association between washout pattern and malignancy. Using SonoVue^®^, Feng et al. reported that 68% of HCCs that showed washout within 120 s after contrast administration were of moderate differentiation, and 82.6% were of poor differentiation [34]. Yao et al. reported that HCC washout <90 s after Sonazoid^®^ administration may be associated with MVI [35] (Table 2).

### 2.3. Computer-Assisted Color Parameter Imaging

Computer-assisted color parameter imaging (CPI) is a new adjunct to CEUS that reconstructs vascular structures by adding temporal information regarding vascular inflow. This post-processing technique is based on the maximum intensity projection method of CEUS in which the arrival time of microbubbles at each part of the lesion is a parameter used to create a color map in any color on the B-mode image [36]. A study of HCC patients who underwent CEUS with CPI before RFA reported that 67.4% of well to moderately differentiated HCCs had a centrifugal perfusion CPI pattern, whereas 84.4% of poorly differentiated or undifferentiated HCCs had a centripetal pattern (*p* < 0.001). Recurrence-free survival (RFS) after RFA was higher in the centrifugal perfusion pattern group than in the centripetal perfusion pattern group (35.0 months vs. 19.0 months, *p* = 0.001) [37].

### 2.4. Combination of Percutaneous Ablation with CEUS

The combination of percutaneous ablation with CEUS can improve treatment outcome in a different manner than the methods for prediction of malignancy and prognostication after treatment of HCC that have been described so far. In this technique, feeding arteries (hepatic artery branches) leading to the HCC are identified by CEUS and then ablated before tumor ablation. If tumor blood flow can be reduced by ablation of the feeding arteries, the margin of RFA can be extended. Li et al. reported that feeding artery ablation for HCC resulted in overall survival (OS) rates of 100% at 1 year, 91.7% at 2 years, and 70.3% at 3 years; in RFS rates of 79.6% at 1 year, 58.1% at 2 years, and 39.0% at 3 years; and in cumulative local tumor progression (LTP) rates of 0.0% at 1 year, 4.2% at 2 years, and 4.2% at 3 years. They considered that feeding artery ablation may improve local tumor control considering that 3-year LTP RFS of conventional RFA without feeding artery ablation ranged from 10% to 38.6% [38]. In my opinion, this ablation technique may improve prognosis after RFA in HCC patients with suspected high-grade malignancy but who are not suitable for surgical resection or liver transplantation.

## 3. Contrast-Enhanced Computed Tomography

### 3.1. Enhancement Pattern in the Arterial Phase

Contrast-enhanced CT (CECT) is a major modality used in the diagnosis of HCC. Previous studies have predicted the prognosis of HCC according to contrast enhancement pattern in the arterial phase of CECT. Kawamura et al. reported that histopathological malignancy differed depending on the enhancement pattern on contrast enhanced CT [39,40]. They classified enhancement patterns into four types. Among them, the type 4 pattern had a high percentage of poorly differentiated HCC (73%) and high recurrence rate after RFA. Nakachi et al. classified tumor enhancement patterns in the arterial phase into without non-enhanced-area and with non-enhanced-area groups, and compared histopathological malignancy between these groups. Tumor biopsies and resection specimens were used to determine histological grade in these groups. The rate of poorly differentiated HCC was 15% in the without non-enhanced group and 85% in the with non-enhanced group (*p* < 0.001). Even in HCCs of 3 cm or less, the rate of poorly differentiated HCC was 25% in the without non-enhanced group and 75% in the with non-enhanced group (*p* < 0.001) [41]. We investigated the association of tumor enhancement patterns of HCC on the arterial phase of CECT with critical recurrence and cancer-related death after RFA. Critical recurrence, which is hard to curatively treat, was defined as “three intrahepatic recurrences, recurrence with vascular invasion, seeding, dissemination, and/or extrahepatic metastasis” [42]. We classified HCC patients treated with RFA into two groups according to the arterial tumor enhancement pattern as the heterogeneous enhancement group (Figure 3) and homogeneous enhancement group (Figure 4). The cumulative 5-year overall recurrence rates in the heterogeneous and homogeneous enhancement groups were 84% and 75%, respectively (*p* = 0.035). The cumulative 5-year critical recurrence rates in the heterogeneous and homogeneous enhancement groups were 42% and 22%, respectively (*p* = 0.005) [42] (Table 3).

### 3.2. Intratumoral Arteries and Enhancement Pattern

In their study based on CECT imaging prior to liver resection, Zhao et al. reported that the presence of intratumoral arteries in the arterial phase (*p* < 0.001), non-nodule type (*p* < 0.001), peritumoral enhancement in the arterial phase (*p* < 0.001), and absence of a radiographic tumor capsule (*p* < 0.001) were strongly associated with MVI [43]. Li et al. investigated the relationship between the presence of a two-trait predictor of venous invasion (TTPVI) on preoperative imaging and postoperative prognosis [44]. The presence of internal arteries in the arterial phase and hypoattenuating halos in the portal venous or delayed phases is defined as TTPVI, and has been reported to correlate with specific HCC molecular profiles derived from microscopic venous invasion gene profiles related to cell proliferation, angiogenesis, and MVI [45]. The 1-, 2-, and 3-year RFS rates in the TTPVI-present group were 54.84%, 37.52%, and 33.67%, and those in the TTPVI-absent group were 81.08%, 69.99%, and 65.54%, respectively (*p* < 0.001). The 1-, 3-, and 5-year OS rates in the TTPVI-present group were 88.50%, 66.39%, and 57.32%, and those in the TTPVI-absent group were 95.92%, 89.57%, and 83.05%, respectively (*p* < 0.001) [44].

## 4. Magnetic Resonance Imaging

MRI has high tissue resolution and can provide information not only on anatomical structure, qualitative diagnosis, and extent of the lesion, but also on the activity of the lesion, and enables evaluation of organ function through contrast administration and diffusion-weighted imaging (DWI).

Gadolinium ethoxybenzyl diethylenetriamine pentaacetic acid (Gd-EOB-DTPA) is used widely as a hepatocyte-specific MRI contrast agent for the detection and diagnosis of liver tumors. Contrast-enhanced MRI with Gd-EOB-DTPA can evaluate the hemodynamics of the liver and liver tumors in the early phase as well as hepatocyte function in the hepatobiliary phase (HBP). The HBP is characterized by enhancement of the hepatic parenchyma by uptake of Gd-EOB-MRI into the hepatocytes. This contrast agent is becoming increasingly important for the detection and diagnosis of HCC [46].

### 4.1. Enhancement Pattern in the Hepatobiliary Phase

Typical hypervascular HCCs show enhancement in the arterial phase and hypointensity in the HBP. However, hepatic nodules can appear hypointense in the HBP without arterial-phase enhancement [47,48]. Such borderline nodules consist of benign hepatocellular nodules or early-stage HCC pathologically. Because HCC progresses in stages from dysplastic nodules to early-stage HCC and finally to typical hypervascular HCC, non-hypervascular hypointense hepatic nodules in the HBP can be considered precursors to hypervascular HCC [49,50,51,52].

In patients with HCC, the presence of such non-hypervascular hypointense hepatic nodules in the HBP has been reported as a significant risk factor for recurrence following HCC treatment [53,54,55,56]. Toyoda et al. reported that the recurrence rate after treatment for early stage HCC was significantly higher in patients with non-hypervascular hypointense nodules than patients without hypointense nodules, regardless of treatment modality (radiofrequency therapy or hepatic resection) [56]. The presence of non-hypervascular hypointense nodules on pretreatment Gd-EOB-DTPA-enhanced MRI was the only factor independently associated with the higher recurrence rate. Furthermore, the presence of non-hypervascular hypointense nodules also tended to be associated with lower survival after treatment in multivariate analysis, but the survival rate in patients with non-hypervascular hypointense nodules was significantly lower than in those without non-hypervascular hypointense nodules. Lee et al. also reported that the presence of non-hypervascular hypointense nodules on preoperative Gd-EOB-DTPA-enhanced MRI was a significant predictor of poor recurrence-free survival after RFA for early stage HCCs. In patients without non-hypervascular hypointense nodules, the estimated 1-, 3-, and 5-year recurrence-free survival rates after RFA were 100%, 78.6%, and 71.3%, respectively, compared to 73.4%, 46.2%, and 27.9%, respectively, in patients with non-hypervascular hypointense nodules [53]. By recurrence type, the cumulative incidence of intrahepatic distant recurrence (IDR) was significantly higher in patients with non-hypervascular hypointense nodules than in those without non-hypervascular hypointense nodules. Such non-hypervascular hypointense nodules may have a poorer prognosis because HCC is more likely to occur in the background liver rather than in HCC present at diagnosis [56]. Moreover, Iwamoto et al. reported that the presence of non-hypervascular hypointense hepatic nodules was an independent risk factor for new intrahepatic recurrence predicting recurrence of hypervascular transformation of non-hypervascular hypointense hepatic nodules. They concluded that patients with non-hypervascular hypointense nodules during the HBP of Gd-EOB-DTPA-enhanced MRI require intensive follow-up after RFA because non-hypervascular hypointense hepatic nodules have potential for hypervascular transformation and their presence is a risk factor for new intrahepatic recurrence [54].

Bae et al. reported prediction of outcome after RFA by analyzing other features in the HBP. They reported that satellite nodules on HBP images were associated with poor disease-free survival (hazard ratio [HR] 5.037; 95% CI, 1.061–23.903) and poor OS (HR 9.398; 95% CI, 1.480–59.668), and that peritumoral hypointensity on HBP images was also associated with poor OS (HR 13.062; 95% CI, 1.627–104.840) [57] (Table 4). These HBP features may be associated with pathological MVI, which is a well-known risk factor for early recurrence and/or poor OS in patients with HCC [58,59,60].

### 4.2. Enhancement Pattern in the Arterial Phase

Several reports have predicted outcome after RFA for HCC by means of extracellular contrast agent-enhanced MR imaging without HBP.

Hu et al. reported that rim enhancement, defined as the presence of ring-like areas of enhancement with central relatively hypointense areas in the arterial phase, was a statistically significant independent predictor of LTP after RFA. The 1-, 2-, and 3-year LTP-free survival rates in patients with rim enhancement were 73.3%, 63.3%, and 60%, respectively, and these rates were significantly lower than those in patients with non-rim enhancement [61]. The reason why patients with rim enhancement are more likely to have LTP after RFA may be that this pattern might indicate poor differentiation, rapid growth, absence of a capsule, infiltrative growth, and microvascular invasion [39,62,63,64,65].

Petukhova-Greenstein et al. reported that continuity of the enhancing “capsule” in the portal venous or delayed phase was associated with poorer progression-free survival (PFS) after RFA for HCC within the Milan criteria [66]. Continuity of an enhancing capsule was classified as the absence, discontinuity, or continuity of a capsule appearance in the portal venous or delayed phase. Patients with an absent, discontinuous, or continuous enhancing “capsule” had median PFS values of 24.0 (95% CI, 18.1–23.7), 17.0 (95% CI, 15.7–19.5), and 13.0 (95% CI, 9.12–18.88) months, respectively. The MRI finding of a portal or delayed-phase enhancing “capsule” may be mirrored by pathologic findings characteristic of HCC, such as peritumoral fibrous tissue and hyperintense tissue represented by vessels and bile ducts compressed by tumor growth [67].

### 4.3. Diffusion-Weighted Imaging

DWI is a non-invasive MRI technique that provides tissue contrast by measuring diffusion of water molecules in tissue without gadolinium contrast material, and is useful for detecting malignant tumors. The apparent diffusion coefficient (ADC) in DWI is a measure of the magnitude of diffusion of water molecules in a tissue. ADC values are automatically calculated by the scanner software and displayed as an ADC map to indicate the degree of diffusion of water molecules through different tissues.

Mori et al. reported that the signal intensity of HCC on the ADC map was strongly associated with outcome after RFA [68]. They defined hypointense HCC as an obviously hypointense tumor compared with surrounding liver on the ADC map (Figure 5) and defined non-hypointense HCC as isointense or hyperintense tumor compared with the surrounding liver (Figure 6). They found that the local recurrence rate after RFA for HCC with more than three tumors or more than 3 cm in diameter was significantly higher in the hypointense group than in the non-hypointense group. The cumulative 2-year local recurrence rates of the hypointense and non-hypointense groups were 18% and 7%, respectively. In addition, the intrahepatic metastasis rate, the recurrence with vascular invasion rate, and the extrahepatic metastasis rate were all higher in the hypointense group than in the non-hypointense group. They concluded that hypointensity on the ADC map was an independent factor related to recurrence and survival after RFA. Hu et al. also reported that the ADC value of HCC was useful for predicting the LTP of HCCs after RFA [61]. As hypointense HCCs on the ADC map are considered to have poor histologic differentiation and a high frequency of microscopic portal vein invasion [69], prognosis after RFA for these HCCs was poor, with a high incidence of advanced as well as local recurrence (Table 5).

## 5. 18F-Fluorodeoxyglucose Positron Emission Tomography

18F-fluorodeoxyglucose positron emission tomography (18F-FDG PET) has emerged as a very effective nuclear medicine imaging tool for obtaining diagnosis, assigning treatment, and assessing tumor response after intervention. Malignant tumor cells overexpress glucose transporter 1 on the plasma membrane, resulting in increased glucose uptake into tumor cells. In addition, hexokinase, a glucose oxidase, is activated in the tumor and delivers glucose to the glycolytic system [70,71]. FDG, which has a structure similar to glucose, is taken up by tumor cells via glucose transporter 1 and is phosphorylated by hexokinase to form FDG-6 phosphate. However, unlike glucose, it cannot proceed through the glycolytic system and accumulates in tumor cells as FDG-6 phosphate. FDG-6 phosphate is dephosphorylated by G-6-Pase and excreted extracellularly because normal hepatocytes are rich in G-6-Pase, which is involved in glycogen production and storage. FDG does not accumulate in well-differentiated HCC because G-6-Pase is present as in normal hepatocytes [72,73]. The diagnostic performance of 18F-FDG PET for naïve HCC is estimated to be 50–70% [74].

### 5.1. Standardized Uptake Value

In PET/CT, the degree of accumulation of radiopharmaceuticals in a lesion is treated as a quantitative value, termed standardized uptake value (SUV), which is the radioactivity concentration measured on the image and corrected for dose and weight. The maximum value per pixel of the lesion is expressed as SUVmax. The possibility of using SUV to determine malignancy in HCC has been reported. Numerous studies have employed SUVmax and the tumor to normal liver SUV ratio (TLR) in patients who underwent surgical resection and liver transplantation [75,76,77,78,79,80,81,82] (Table 6); however, the cutoff values for SUVmax and TLR vary among studies. It is difficult to set an optimal cutoff value because the SUV varies depending on factors such as the imaging model, blood glucose level at the time of examination, image reconstruction method, and liver function [83,84].

### 5.2. 18F-Fluorodeoxyglucose Uptake

In contrast, it is easy to visually assess the presence or absence of FDG uptake in HCC. Ida et al. visually evaluated FDG accumulation in HCC and prospectively examined prognosis after RFA. The degree of 18F-FDG uptake in a nodule seen on 18F-FDG-PET was compared visually with that in surrounding liver. Tumors with stronger 18F-FDG uptake, either as a whole or partially, in comparison with surrounding liver were termed PET-positive (Figure 7), and those with 18F-FDG uptake equal to that of surrounding liver were termed PET-negative (Figure 8). The cumulative 1-year RFS rates in the PET-positive and PET-negative groups were 30% and 64%, respectively (*p* = 0.017). The cumulative 1-year 3 or more recurrences rates of the PET-positive and PET-negative groups were 36% and 6%, respectively (*p* < 0.001). The cumulative 5-year survival rates in the PET-positive and PET-negative groups were 22% and 88%, respectively (*p* < 0.001). In multivariate analysis, 18F-FDG PET positivity was an independent factor for recurrence and survival after RFA [85].

Visual assessment of 18F-FDG uptake has also been shown to be useful in predicting prognosis after treatment in patients who have undergone liver resection or transplantation [86,87,88,89,90]. Lim et al. reported the relationship between preoperative PET/CT findings and histopathological results, and between preoperative PET/CT and post-liver-resection recurrence. MVI was observed in 53% of the PET-positive group and in 21% of the PET-negative group (*p* = 0.003). Based on these results, they concluded that PET/CT can predict the presence of MVI (OR = 3.4, 95% CI 1.1–10.3; *p* = 0.03). The recurrence rate within 24 months after resection was 39% in the PET-positive group and 17% in the PET-negative group (*p* = 0.03) [87]. Lee et al. found that, within the Milan criteria, the RFS rates in the PET-positive group were 91.3% at 1 year, 76.3% at 3 years, and 76.3% at 5 years, whereas those in the PET-negative group were 97.2% at 1 year, 94.8% at 3 years, and 92.3% at 5 years (*p* = 0.031). However, no significant difference in OS rates was found between the groups (*p* = 0.140). On the other hand, beyond the Milan criteria, the RFS rates and OS rates were significantly lower in the PET-positive group than the PET-negative group (*p* < 0.001) [90] (Table 7).

## 6. Limitation

Tumor biopsy is the gold standard for diagnosing the degree of differentiation of malignant tumor. However, percutaneous ablation for HCC is performed without preoperative tumor biopsy because of risks such as dissemination. Therefore, it is difficult to directly compare a patient’s imaging-determined malignancy with the actual histological grade. 

## 7. Conclusions

Previously, tumor size, number, and location have been important factors in the selection of treatment for HCC. However, as mentioned above, various imaging modalities have recently been used to predict malignancy of HCC prior to percutaneous ablation. If the prognosis is predicted to be poor by imaging before ablation, a careful decision must be made as to whether ablation should be the first choice, even if HCC is within the indication for ablation. Furthermore, in recent years, attempts have been made in the field of chemotherapy to predict response rates using imaging studies. In the future, prediction of malignancy by imaging diagnosis will become more important factor.

## Figures and Tables

**Figure 1 diagnostics-13-03058-f001:**
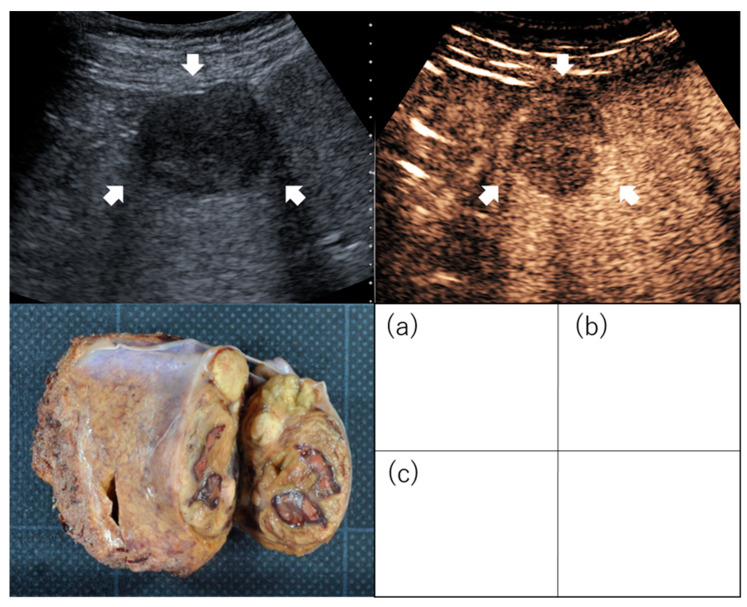
Images of irregularly shaped hepatocellular carcinoma on contrast-enhanced ultrasound. Images of confluent multinodular type: Hepatocellular carcinoma (3 cm, segment 4) was evaluated as an irregular type by conventional B-mode ultrasound (**a**) and as irregular type in the post-vascular phase of contrast-enhanced ultrasound (**b**). Examination of the resected specimen identified the mass as confluent multinodular type (**c**). Arrows indicate the tumor.

**Figure 2 diagnostics-13-03058-f002:**
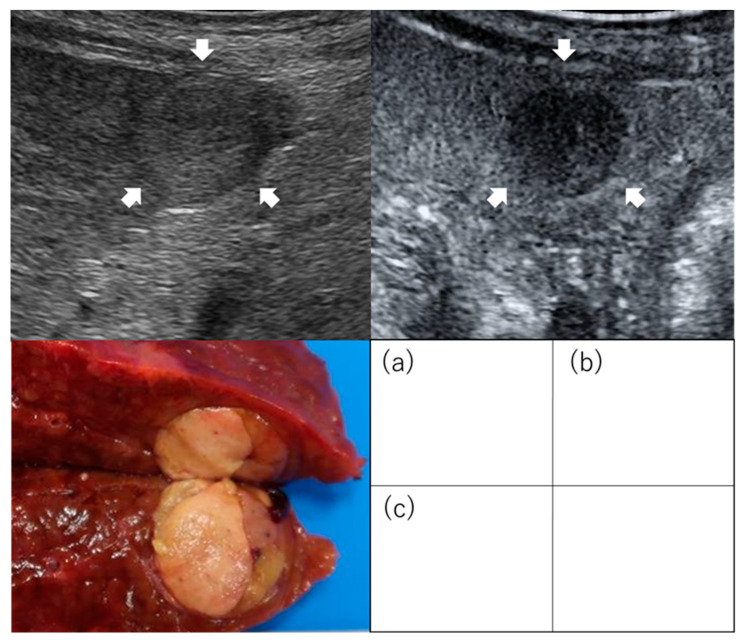
Images of regularly shaped hepatocellular carcinoma on contrast-enhanced ultrasound. Images of simple nodular type: (**a**) Hepatocellular carcinoma (2 cm, segment 3) was evaluated as non-irregular type by conventional B-mode ultrasound (**b**) and non-irregular type as an image defect in the post-vascular phase of contrast-enhanced ultrasound. (**c**) Examination of the resected specimen identified the mass as the simple nodular type. Arrows indicate the tumor.

**Figure 3 diagnostics-13-03058-f003:**
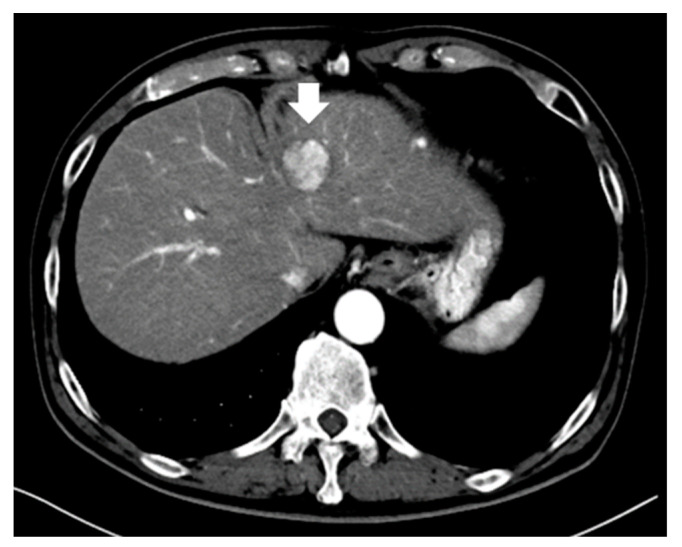
A typical small hepatocellular carcinoma (3.0 cm) showing arterial heterogeneous tumor enhancement on contrast-enhanced computed tomography (arrow indicates tumor).

**Figure 4 diagnostics-13-03058-f004:**
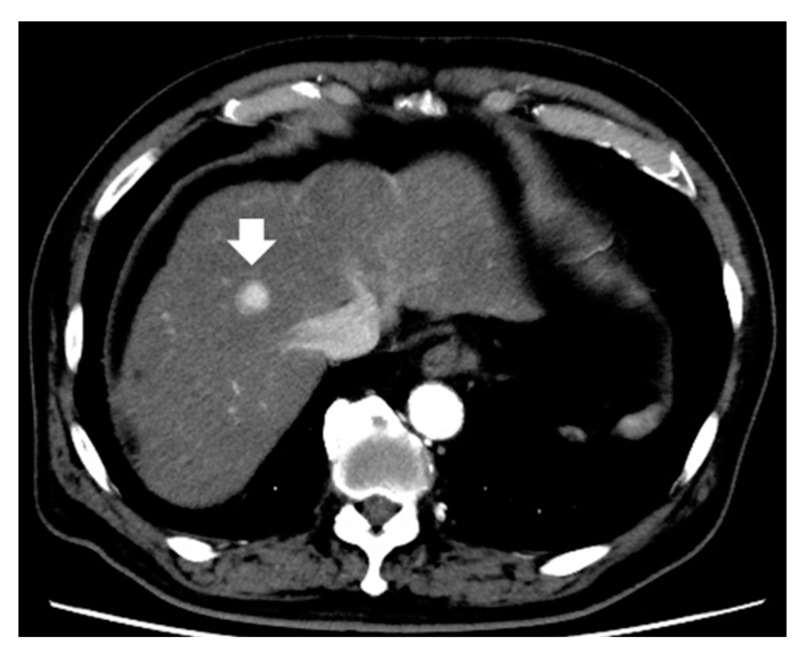
Representative small hepatocellular carcinoma (2.0 cm) shows arterial homogeneous tumor enhancement on contrast-enhanced computed tomography (arrow indicates tumor).

**Figure 5 diagnostics-13-03058-f005:**
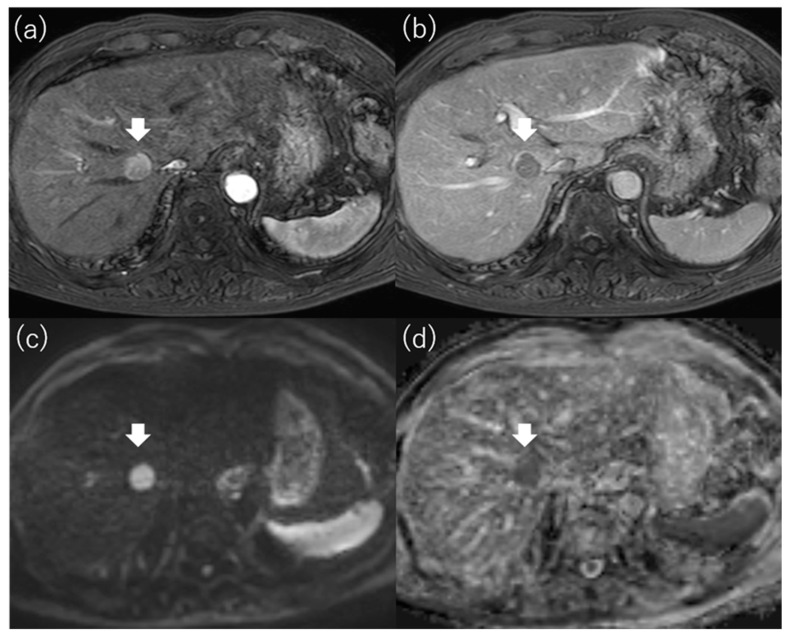
Images of hypointense hepatocellular carcinoma on apparent diffusion coefficient map. Gd-EOB-DTPA-enhanced magnetic resonance imaging reveals a 1.7 cm hepatocellular carcinoma in segment 8 that shows staining in the arterial phase (**a**) and washout in the portal venous phase (**b**). The tumor is hyperintense on diffusion-weighted imaging with a b-factor of 1000 (**c**) and hypointense on the apparent diffusion coefficient map (**d**). Gd-EOB-DTPA, gadolinium ethoxybenzyl diethylenetriamine pentaacetic acid.

**Figure 6 diagnostics-13-03058-f006:**
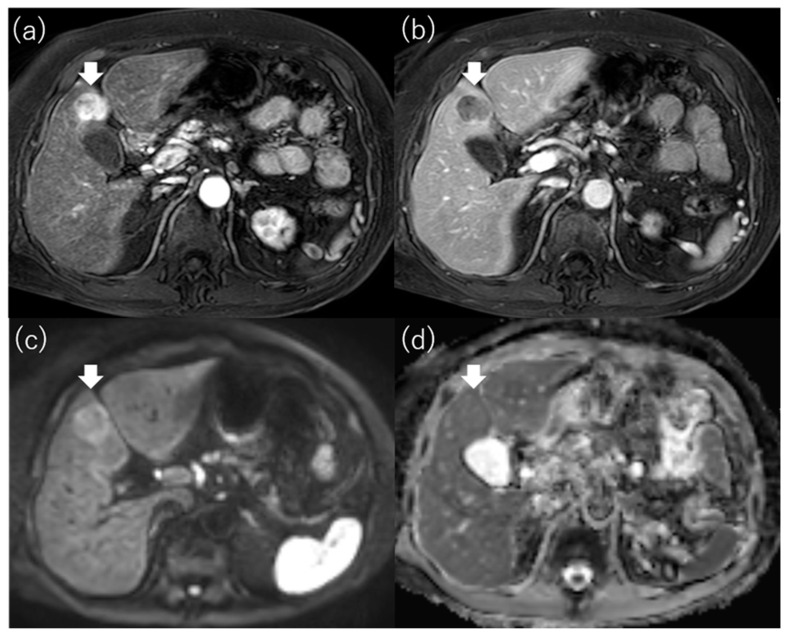
MR images of non-hypointense hepatocellular carcinoma on apparent diffusion coefficient map. Gd-EOB-DTPA-enhanced magnetic resonance imaging reveals a 2.6 cm hepatocellular carcinoma in segment 4 that shows staining in the arterial phase (**a**) and washout in the portal venous phase (**b**). The tumor is hyperintense on diffusion-weighted imaging with a b-factor of 1000 (**c**) and hypointense on the apparent diffusion coefficient map (**d**). Gd-EOB-DTPA, gadolinium ethoxybenzyl diethylenetriamine pentaacetic acid.

**Figure 7 diagnostics-13-03058-f007:**
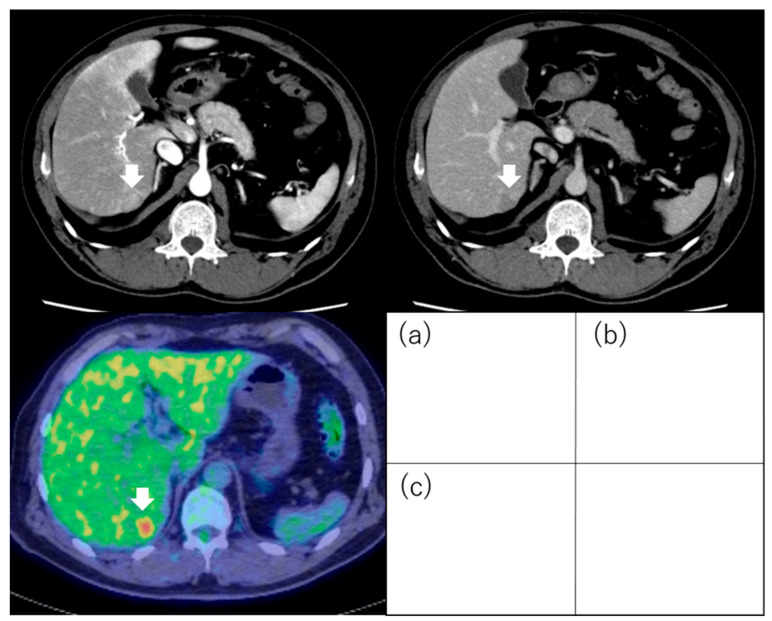
Positron emission tomography images of hepatocellular carcinoma following 18F-fluorodeoxyglucose uptake. Hepatocellular carcinoma (1.8 cm, segment 6) with positive 18F-fluorodeoxyglucose uptake on positron emission tomography: The tumor exhibits staining during the arterial phase of contrast computed tomography (**a**) and washout during the equilibrium phase (**b**). The tumor has higher 18F-fluorodeoxyglucose uptake than that of the surrounding liver on positron emission tomography (**c**). Arrows indicate the tumor.

**Figure 8 diagnostics-13-03058-f008:**
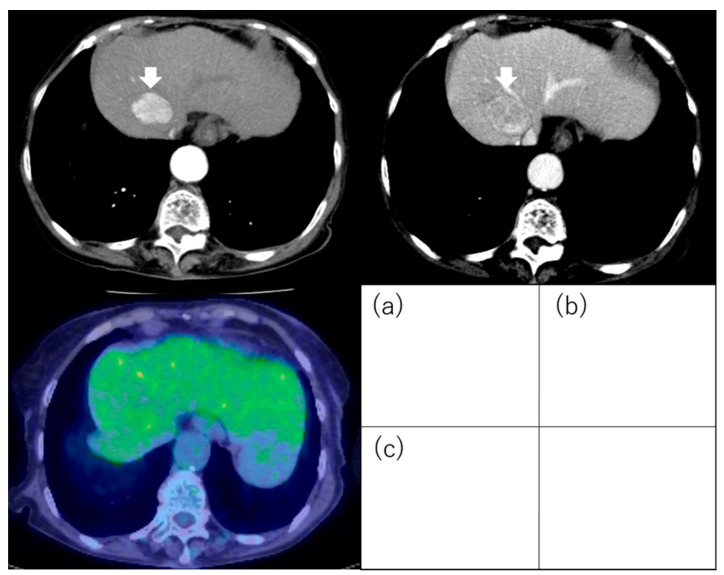
Hepatocellular carcinoma with negative 18F-fluorodeoxyglucose uptake on positron emission tomography. The hepatocellular carcinoma (3.5 cm, segment 4) exhibits staining during the arterial phase of contrast computed tomography (**a**) and washout during the equilibrium phase (**b**). The tumor has equal 18F-fluorodeoxyglucose uptake to that of the surrounding liver on positron emission tomography (**c**). Arrows indicate the tumor.

**Table 1 diagnostics-13-03058-t001:** Gross classification with post-vascular phase image of contrast-enhanced ultrasonography.

Study	N	Inclusion Criteria of Tumor	Treatment	Evaluation	Results
Hatanaka et al. [28]	29	Solitary and ≤5 cm ≤3 tumors and ≤3 cm	LR	Gross type (SN type or non-SN type)	Sensitivity, 96%; specificity, 80%; accuracy, 90%
Li et al. [29]	31	Early stage	LR	Microvascular invasion	Non-SN type in the post-vascular phase image was an independent predictor of microvascular invasion (OR, 30.51; 95% CI, 2.335–398.731, *p* = 0.009). Sensitivity, 93.3%; specificity, 81.3%; positive predictive value, 82.4%; negative predictive value, 92.9%
Nuta et al. [30]	73	Solitary and ≤5 cm	LR	Gross type (SN type or non-SN type) Outcome (recurrence)	In the post-vascular phase, predictability for high-grade malignant potential was as follows: sensitivity was 93%, specificity was 85%, positive predictive value was 97%, negative predictive value was 73%, and accuracy was 92%. Irregular defect pattern was one of the independent factors for metastatic recurrence (HR, 4.388; 95% CI, 1.008–19.089; *p* = 0.049).

CI: confidence interval; HR: hazard ratio; LR: liver resection; RFA: radiofrequency ablation; OR: odds ratio; SN: simple nodular.

**Table 2 diagnostics-13-03058-t002:** Malignant potential with intratumoral blood flow in the arterial phase or the portal phase of contrast-enhanced ultrasonography.

Study	*n*	Inclusion Criteria of Tumor	Treatment	Evaluation	Results
Maruyama et al. [32]	54	Solitary and ≤5 cm ≤3 tumors and ≤3 cm	RFA	Outcome (Recurrence)	Intensity differences at the early arterial time is one of the independent factors for intrahepatic distant recurrence (HR, 2.7; 95% CI, 1.2–5.8; *p* = 0.014).
Han et al. [33]	125	Solitary and 5 cm or less ≤3 tumors and ≤3 cm	RFA	Outcome (Recurrence)	Tumor peak intensity was a significant independent risk factor for intrahepatic recurrence after RFA (HR, 0.3; 95% CI, 0.1–0.9).
Feng et al. [34]	271	Early stage	LR	Differentiation	Washout before 120 s from contrast injection group had poorly differentiation. Sensitivity, 98.0%; specificity, 77.8%; positive predictive value, 96.0%; negative predictive value, 48.8%
Yao et al. [35]	211	Early stage	LR	Microvascular invasion	Washout time (<90 s) (OR: 2.755, 95% CI: 1.227–6.187) was one of the independent predictors of microvascular invasion.

CI: confidence interval; HR: hazard ratio; LR: liver resection; RFA: radiofrequency ablation; OR: odds ratio; SN: simple nodular.

**Table 3 diagnostics-13-03058-t003:** Malignant potential with enhancement pattern in the arterial phase of contrast-enhanced computed tomography.

Study	*n*	Inclusion Criteria of Tumor	Treatment (n)	Evaluation	Results
Kawamura et al. [40]	191	Solitary and ≤3 cm	Resection (60) RFA (131)	Outcome (recurrence)	The type 4 enhancement pattern was an independent factor for recurrence (HR, 27.68; 95% CI, 6.82–112.33; *p* < 0.001).
Nakachi et al. [41]	223	Early stage	Biopsy Resection	Differentiation	HCC with enhancement with non-enhanced area predicted poorly differentiation. Predictive value as follows; sensitivity was 85%, specificity was 76%, positive predictive value was 93%, negative predictive value was 97%, and accuracy was 77%.
Shimizu et al. [42]	226	≤3 tumors and ≤3 cm	RFA	Outcome (recurrence, OS)	Heterogeneous enhancement pattern was one of the independent factors for critical recurrence (HR, 2.753; 95% CI, 1.453–5.219; *p* = 0.002) and related cancer death (HR, 2.369; 95% CI, 1.187–4.726; *p* = 0.014).

CI: confidence interval; HCC: hepatocellular carcinoma; HR: hazard ratio; OS: overall survival; RFA: radiofrequency ablation.

**Table 4 diagnostics-13-03058-t004:** Malignant potential with enhancement pattern in the hepatobiliary phase of magnetic resonance imaging.

Study	*n*	Inclusion Criteria of Tumor	Treatment (n)	Evaluation	Results
Lee et al. [53]	345	Solitary and ≤3 cm	LR (123) RFA (222)	Outcome (recurrence)	In RFA group, non-hypervascular hypointense nodule was one of the independent factors for recurrence (HR, 1.78; 95% CI, 1.20–2.63; *p* = 0.004).
Iwamoto et al. [54]	91	Early stage	RFA	Outcome (recurrence)	Non-hypervascular hypointense nodule was one of the independent factors for intrahepatic distant recurrence (HR, 4.37; 95% CI, 2.13–8.86; *p* < 0.01).
Toyoda et al. [55]	77	Early stage	LR	Outcome (recurrence)	Non-hypervascular hypointense nodule was an independent factor for multicentric recurrence (HR, 2.84; 95% CI, 1.69–4.84; *p* = 0.0002).
Toyoda et al. [56]	138	BCLC stage 0 or A	LR (76) RFA (62)	Outcome (recurrence, OS)	Non-hypervascular hypointense nodule was an independent factor for recurrence (HR, 1.68; 95% CI, 1.26–2.25; *p* = 0.0005) and which was one of the independent factors for OS (HR, 1.63; 95% CI, 0.99–2.81; *p* = 0.05).
Bae et al. [57]	183	BCLC stage 0 or A	LR (61) RFA (61) TACE (61)	Outcome (recurrence, OS)	In RFA group, existing satellite nodules was one of the independent factors for disease-free survival (HR, 5.04; 95% CI, 1.06–23.90; *p* = 0.04); moreover, peritumoral hypointensity (HR, 13.06; 95% CI, 1.63–104.84; *p* = 0.02) and existing satellite nodules (HR, 9.40; 95% CI, 1.48–59.67; *p* = 0.02) on HBP were related to OS.

BCLC: Barcelona Clinic Liver Cancer; CI: confidence interval; HBP: hepatobiliary phase; HR: hazard ratio; LR: liver resection; LTP: liver transplantation; OS: overall survival; RFA: radiofrequency ablation; TACE: transcatheter arterial chemoembolization.

**Table 5 diagnostics-13-03058-t005:** Malignant potential with signal intensity on the ADC map of magnetic resonance imaging.

Study	*n*	Inclusion Criteria of Tumor	Treatment (n)	Evaluation	Results
Mori et al. [68]	136	≤3 tumors and ≤3 cm	RFA	Outcome (recurrence, OS)	Hypointensity on the ADC map was one of the independent factors for recurrence (HR, 2.651; 95% CI, 1.530–4.593; *p* = 0.001), local recurrence (HR, 5.602; 95% CI, 1.526–20.568; *p* = 0.009), critical recurrence (HR, 2.555; 95% CI, 1.171–5.571; *p* = 0.018), and survival (HR, 2.945; 95% CI, 1.124–7.721; *p* = 0.028).
Mori et al. [69]	52	Solitary and ≤5 cm	LR	Outcome (recurrence)	Hypointensity on the ADC map was one of the independent factors for metastatic recurrence (HR, 12.279; 95% CI, 1.486–101.48; *p* = 0.020).

ADC: apparent diffusion coefficient; CI: confidence interval; HR: hazard ratio; LR: liver resection; OS: overall survival; RFA: radiofrequency ablation.

**Table 6 diagnostics-13-03058-t006:** Malignant potential with SUV value of 18F-fluorodeoxyglucose positron emission tomography.

Study	*n*	Inclusion Criteria of Tumor	Treatment (n)	Point of View	Evaluation	Results
Lee et al. [75]	59	Milan criteria (within/beyond)	LTP	TSUVmax/LSUVmax (cutoff value; 1.15)	Outcome (recurrence)	Significantly difference was seen about recurrence-free survival. TSUVmax/LSUVmax < 1.15; the 1 year and 2 year recurrence-free survival rate was 97% and 97%, respectively. TSUVmax/LSUVmax ≥ 1.15; the 1 year and 2 year recurrence-free survival rate was 57% and 42%, respectively.
Detry et al. [76]	27	Milan criteria (within/beyond)	LTP	TSUVmax/LSUVmax (cutoff value; 1.15)	Outcome (recurrence, OS)	TSUVmax/LSUVmax > 1.15 was one of the independent factors for recurrence-free survival (HR, 14.4; *p* = 0.01) and overall survival (HR, 5.62; *p* = 0.04).
Hong et al. [77]	123	Milan criteria (within/beyond)	LTP	TSUVmax/LSUVmax (cutoff value; 1.10)	Outcome (recurrence, OS)	TSUVmax/LSUVmax > 1.10 was an independent factor for recurrence (HR, 9.776; 95% CI, 3.557–26.816; *p* < 0.001).
HSU et al. [78]	147	“Solitary and ≤6.5 cm” or “≤3 tumors and ≤4.5 cm” and total tumor diameter ≤8 cm	LTP	TNR (cutoff value; 2.0)	Outcome (recurrence)	TNR ≥ 2.0 was one of the independent factors for recurrence (HR, 13.52; 95% CI, 4.77–38.29; *p* < 0.001).
Hyun et al. [79]	317	BCLC stage 0 or A	Curative treatment (LR or RFA or LTP) (195) TACE (122)	TLR (cutoff value; 2.0)	Outcome (recurrence, OS)	TLR ≥ 2.0 was an independent factor for OS in curative treatment group (HR, 2.68; 95% CI, 1.16–6.15; *p* = 0.020).
Cho et al. [80]	56	Early stage	LR	SUVmax (cutoff value; 4.9)	Outcome (recurrence, OS)	SUVmax was not significant independent factor for recurrence (*p* = 0.262) and OS (*p* = 0.717).
Hyun et al. [81]	158	BCLC stage 0 or A	LR	TLR (cutoff value; 1.3)	Microvascular invasion Outcome (recurrence)	Predictability of TLR for microvascular invasion as follows: sensitivity was 85.5%, specificity was 54.9%, positive predictive value was 63.7%, negative predictive value was 80.4%. TLR was one of the independent factors for metastatic recurrence (HR, 2.43; 95% CI, 1.01–5.84; *p* = 0.047).
Yoh et al. [82]	207	Solitary	LR	TNR (cutoff value; 2.0)	Outcome (recurrence, OS)	TNR was one of the independent factors for OS (HR, 1.743; 95% CI, 1.114–2.648; *p* = 0.016).

CI: confidence interval; HR: hazard ratio; LR: liver resection; LSUV: liver SUV; LTP: liver transplantation; OS: overall survival; PET: positron emission tomography; RFA: radiofrequency ablation; SUV: standardized uptake values; TLR: tumor to normal liver SUV ratio; TNR: tumor to nontumor ratio; TSUV: tumor SUV.

**Table 7 diagnostics-13-03058-t007:** Malignant potential with PET uptake (positive or negative) of 18F-fluorodeoxyglucose positron emission tomography.

Study	*n*	Inclusion Criteria of Tumor	Treatment (n)	Evaluation	
Ida et al. [85]	121	≤3 tumors and ≤3 cm	RFA	Outcome (recurrence, OS)	PET positivity was one of the independent factors for metastatic recurrence (HR, 10.297; 95% CI, 3.128–33.898; *p* < 0.001) and OS (HR, 7.300; 95% CI, 1.920–27.751; *p* = 0.004).
Park et al. [86]	92	Early stage	LR	Outcome (recurrence, OS)	PET positivity was one of the independent factors for disease-free survival (HR, 2.8; 95% CI, 1.273–6.158; *p* = 0.01). In PET-positive HCC, OS of narrow resection margin group was significantly shorter than that of wide resection margin group (*p* < 0.001).
Lim et al. [87]	78	Early stage	LR	Microvascular invasion Outcome (recurrence)	Predictability of PET for microvascular invasion of as follows: sensitivity was 62%, specificity was 76%, positive predictive value was 71%, and negative predictive value was 53%, respectively. PET positive was one of the independent factors for early recurrence (HR, 5.8; 95% CI, 1.6–20.4; *p* = 0.006).
Yang et al. [88]	38	Milan criteria (within/beyond)	LTP	Outcome (recurrence)	The association between PET positivity and tumor recurrence was significant (*p* = 0.003). The 2-year recurrence-free survival rates for the PET-positive group and PET-negative group were 46.1% and 85.1%, respectively (*p* = 0.0005).
Kornberg et al. [89]	42	Milan criteria (within/beyond)	LTP	Microvascular invasion Outcome (recurrence)	PET positivity was an independent factor for microvascular invasion (HR, 13.4; 95% CI, 0.003–0.126; *p* = 0.001). The 3-year recurrence-free survival rates beyond the Milan criteria/PET-negative group and beyond the Milan criteria/PET-positive group were 94% and 29%, respectively (*p* < 0.001).
Lee et al. [90]	280	Milan criteria (within/beyond)	LTP	Outcome (recurrence, OS)	In patients beyond the Milan criteria, PET positivity was one of the independent factors for recurrence-free survival (HR, 3.803; 95% CI, 1.876–7.707; *p* < 0.001) and OS (HR, 2.714; 95% CI, 1.239–5.948; *p* = 0.013).

CI: confidence interval; HCC; hepatocellular carcinoma; HR: hazard ratio; LR: liver resection; LTP: liver transplantation; OS: overall survival; PET: positron emission tomography; RFA: radiofrequency ablation; SUV: standardized uptake values.

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
