# Peer review of "Predicting Outcome after Percutaneous Ablation for Early-Stage Hepatocellular Carcinoma Using Various Imaging Modalities"

_diagnostics, 2023, doi:10.3390/diagnostics13193058_

Round 1

Reviewer 1 Report

The paper from Shimizu and coworkers reports available evidence on the usefulness of different imaging techniques such as US, CT, MRI and 18F-FDG PET for predicting outcome after percutaneous ablation for HCC. The paper represents a complete review of available data from published trials and it is very interesting.

Some recommendations/suggestions:

A) the text is very full. I think that abbreviating some sentences or explanations and using practical charts, lists or tables to get the concise findings of the manuscript in a glance, should be very useful. These are recommendations/suggestions for paraghaphs 2,3,4,5. For example:

  1) From Ln 67 to ln 72 could be written as follows: "The gross classification divides HCCs into five types [24]: 

-small nodular type with indistinct margin;

- simple nodular (SN) type, defined as a tumor with a clear round shape;

- single nodular with extranodular growth (SNEG) type, defined as a tumor with a clear round shape similar to the SN type, but with one or more coexisting peri-nodular tumor growths;

- confluent multinodular (CMN) type, defined as tumor comprising a cluster of small nodules.

- infiltrative type”.

  2) The different studies should be insert in tables in which data as sensitivity, specificity, positive predictive value, and negative predictive value, recurrence-free survival rate, findings, etc could be more schematic.

B) Discuss any limitations of available evidence included in the review.

C) The style of letters in Fig. 1, Fig. 2, Fig. 7 and Fig. 8 should be improved (for example the style of Fig 5 and 6 is better);

D) The definitions of abbreviations should be insert only for the first time that are used in the text.

Author Response

Below are our responses to the reviewer’s comments (Manuscript ID
diagnostics-2559476)

Reviewer 1

Comments and Suggestions for Authors

A) The text is very full. I think that abbreviating some sentences or explanations and using practical charts, lists or tables to get the concise findings of the manuscript in a glance, should be very useful. These are recommendations/suggestions for paraghaphs 2,3,4,5.

→I fixed them and shaded in yellow (Page 2, Line 27. Page 3, Line 5).

The different studies should be insert in tables in which data as sensitivity, specificity, positive predictive value, and negative predictive value, recurrence-free survival rate, findings, etc could be more schematic.

→I made tables (including criteria of HCC, results, and so on) about our references (Page 6/8/12/16/20/22/23/27/28).

B) Discuss any limitations of available evidence included in the review.

→I wrote limitation about my report in page 29.

C) The style of letters in Fig. 1, Fig. 2, Fig. 7 and Fig. 8 should be improved (for example the style of Fig 5 and 6 is better);

→I fixed them and shaded in yellow (Page 4/5/24/25).

D) The definitions of abbreviations should be insert only for the first time that are used in the text.

→ I fixed them and shaded in yellow (Page2, Line 13/22. Page 3, Line 14. Page 9, Line 10/20/22/32. Page 14, Line 34. Page 17, Line 14).

Reviewer 2 Report

The present manuscript is a review article comparing different modalities of imaging diagnostics in predicting post ablation/transplantation/resection recurrence. Authors did a great effort. The subject is controversial and is a topic of interest to hepatologists, oncologists, surgeons and radiologists. 

Few suggestions:

1) References should be more recent. Yes, there are few references in 2022 and 2 in 2023; but in such a topic there are a lot escalation in our knowledge in this field. Many references are present to support this manuscript and make it more cited.

2) Images need editing and are not the best resolution. Additionally, the source of the images should be stated. 

3) Through out the manuscript; it is better to put subsections or titles instead of bulky paragraphs and pages.

Good quality.

Author Response

Below are our responses to the reviewer’s comments (Manuscript ID
diagnostics-2559476)

Reviewer 2

Comments and Suggestions for Authors

1) References should be more recent. Yes, there are few references in 2022 and 2 in 2023; but in such a topic there are a lot escalation in our knowledge in this field. Many references are present to support this manuscript and make it more cited.

→To the best our knowledge, there is not much latest literature on the prognostic ability of imaging studies in percutaneous ablation therapy for HCC. Therefore, I presented the literature on the relationship between liver resection or liver transplantation and imaging studies to complement the evidence in ablation therapy,

2) Images need editing and are not the best resolution. Additionally, the source of the images should be stated.

→These images are original in our department and the best one.

3) Through out the manuscript; it is better to put subsections or titles instead of bulky paragraphs and pages.

→I added heading and shaded in yellow (Page 3/7/9/13/17/21/24/29).

Reviewer 3 Report

This is a well-written review on an important topic. 

However, the author should clearly state for the results of all analyzed studies the stage (diameter and number) of HCC, weather the studies are about imaging or treatment. Moreover, it is not clear if the analysis and conclusion are addressed to early stage HCC (if so, the authors should add in the title 'early stage' in front of HCC), or also HCC beyond early stages. If the latter is true, they should comment if, based on the absence of poor prognostic criteria identified at imaging, in selected cases tumor ablation may safely go beyond early stage with similar results as for early stage.

Author Response

Below are our responses to the reviewer’s comments (Manuscript ID
diagnostics-2559476)

Reviewer 3

Comments and Suggestions for Authors

However, the author should clearly state for the results of all analyzed studies the stage (diameter and number) of HCC, weather the studies are about imaging or treatment. Moreover, it is not clear if the analysis and conclusion are addressed to early stage HCC (if so, the authors should add in the title 'early stage' in front of HCC), or also HCC beyond early stages. If the latter is true, they should comment if, based on the absence of poor prognostic criteria identified at imaging, in selected cases tumor ablation may safely go beyond early stage with similar results as for early stage.

→I inserted "early stage" in the title and shaded in yellow. "Predicting outcome after percutaneous ablation for early stage hepatocellular carcinoma using various imaging modalities"

→I made tables (including criteria of HCC, results, and so on) about our references (Page 6/8/12/16/20/22/23/27/28).

Round 2

Reviewer 1 Report

Authors followed tips and suggestions.